 # FedPissa: Towards Federated Personalized Adaptation of Foundation Models via LoRA Subspace Mapping

Wenwen He [1]   Wenke Huang [2]   Yi Liu [3]   Jian Liang [4]   Xirui Li [5]   Guansong Pang [6]   Mang Ye [4]

## Abstract

LoRA efficiently adapts large pre-trained models via low-rank updates, making it a strong parameter-efficient fine-tuning (PEFT) method. When integrated with Federated Learning (FL), it enables collaborative fine-tuning across distributed clients, leveraging rich downstream data without exposing private information. However, this strategy is hindered by data heterogeneity and limits personalization performance. To address this, personalized FedLoRA approaches have been proposed and employ a dual-LoRA architecture, *i.e.*, one branch for global knowledge and another for client-specific adaptation. Nevertheless, this dual-LoRA design introduces additional computational overhead and structural redundancy. To address this limitation, we propose **FedPissa**, the first framework that rethinks single-LoRA via *selective aggregation* and *subspace decorrelation*. We selectively aggregate LoRA components based on their aggregation dynamics, and further apply a decorrelated subspace projection to mitigate heterogeneous update conflicts, reducing cross-client interference and improving personalized adaptation. Experiments on texual and visual scenario show that FedPissa not only achieves up to 35% lower communication and computation cost, but also improves superior compared to counterparts.

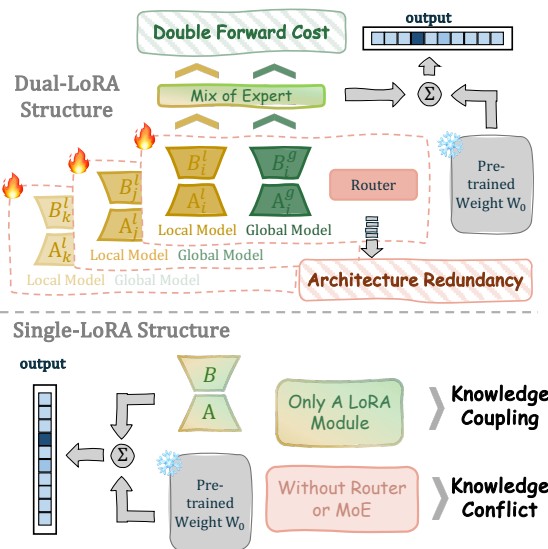

*Figure 1.* **Motivation**. Existing pFedLoRA adopts a **dual-LoRA structure**. Each client maintains a **local** LoRA for personalization and a **global** LoRA for shared knowledge.

## 1. Introduction

Foundation models have demonstrated remarkable generalization and transfer capabilities across a various tasks (Touvron et al., 2023; Vavekanand & Sam, 2024; Jaech et al., 2024; Team et al., 2025; Guo et al., 2025a; Wen et al., 2025; Dosovitskiy et al., 2021). To efficiently adapt these large foundation models to diverse downstream applications, Parameter-Efficient Fine-Tuning (PEFT) methods (Han et al., 2024; Zhou et al., 2024; Zhang et al., 2025) have been widely adopted. Among them, Low-Rank Adaptation (LoRA) (Hu et al., 2022; Hayou et al., 2024; Liu et al., 2024b; Kopiczko et al., 2023) has gained particular attention for its ability to fine-tune models effectively through lightweight low-rank adapters. LoRA injects two trainable low-rank matrices, $A$ for down-projection and $B$ for up-projection, into the frozen model weights, enabling efficient task adaptation while maintaining the expressive power of original model. Although foundation models contain extensive general knowledge across domains, collecting rich task-specific data while preserving privacy remains a major bottleneck for LoRA (Zhang et al., 2024a; Hao et al., 2024).

Work done by Wenke Huang during internship at Tencent. [1]School of National Cyber Security, Wuhan University [2]College of Computing and Data Science, Nanyang Technological University [3]City University of Hong Kong [4]School of Computer Science, Wuhan University [5]Tencent [6]Singapore Management University. Correspondence to: Mang Ye <yemang@whu.edu.cn>.

*Proceedings of the 43$^{rd}$ International Conference on Machine Learning*, Seoul, South Korea. PMLR 306, 2026. Copyright 2026 by the author(s).

To address this challenge, researchers combine LoRA with federated learning (Zhang et al., 2024a) (FedLoRA), allowing clients to fine-tune collaboratively on local data without sharing raw samples (Huang et al., 2026). In FedLoRA, the server aggregates client LoRA updates (e.g., FedAvg (McMahan et al., 2017)), but heterogeneity often makes a single global LoRA poorly fit diverse client distributions. Meanwhile, many personalized FL methods (Shamsian et al., 2021; Ma et al., 2022; Yang et al., 2023; Li et al., 2023) assume full-model updates or classifier-dependent designs, which are not directly compatible with LoRA. This motivates personalized federated LoRA (pFedLoRA) (Yi et al., 2023; Qi et al., 2024), which learns client-specific LoRA modules to better match local data.

Existing pFedLoRA methods fall into two categories. The first uses *static coordination* (Qi et al., 2024; Yang et al., 2024; Guo et al., 2025b), where server and client updates alternate in a fixed schedule, *e.g.*, FedDPA (Yang et al., 2024) alternates between global aggregation and local specialization. The second employs *dynamic clustering* (Zhang et al., 2024b; Wang et al., 2025; Bian et al., 2025), using routing or gating mechanisms to group clients by representation similarity and assign cluster-specific LoRA experts, such as FedLEASE (Wang et al., 2025), which adapts experts based on learned client affinities. Both of them adopt a dual-LoRA architecture framework (as shown in Figure 1), with one LoRA module capturing global knowledge and the other preserving client-specific information. They achieve personalization ats cost of doubled computational overhead and structural redundancy, resulting a burden for each client.

This motivated us to explore an efficient federated adaptation framework for foundation models that uses only a single LoRA module (Guo et al., 2025b; Sun et al., 2024; Hao et al., 2024), eliminating the need for dual-LoRA redundancy. In this paper, we expect to achieve two goals: (1) low computational cost, and (2) strong personalization performance. Unlike conventional pFedLoRA methods, it enables effective separation of global and local knowledge within a streamlined architecture, preserving aggregation quality while enhancing efficiency. However, realizing this design poses two key challenges:

- **C1: Decoupling Global and Personalized Knowledge in a Single LoRA.** A key challenge is to separate globally shared patterns from client-specific knowledge within a *single* LoRA, without dual-branch designs. Existing pFedLoRA methods use distinct modules for global vs. local adaptation to enable explicit coordination (Wang et al., 2025; Yang et al., 2024; Qi et al., 2024). In our setting, we must instead identify which LoRA components should be aggregated for stable collaboration and which should remain local for personalization.
- **C2: Enabling Personalized Adaptation without Dual Branches or Auxiliary Networks.** Many prior methods

rely on auxiliary mechanisms (Masoudnia & Ebrahimpour, 2014; Dou et al., 2024; Liu et al., 2024a; Chen et al., 2024; Li et al., 2021; Zhang et al., 2022) to route or reweight dual LoRA branches for personalization. Such designs are incompatible with a lightweight single-LoRA framework, and removing them often weakens fine-grained control over client-specific adaptation under heterogeneous data distributions across clients.

To address these challenges, we propose FedPissa, the first framework that rethinks single-LoRA through selective aggregation and subspace decorrelation. Our key insight is that $A$ and $B$ play asymmetric roles in LoRA, enabling matrix-level personalization rather than full-LoRA personalization. Empirically, $B$ converges faster and varies more across clients, suggesting it is better for cross-client sharing. Driven by this insight, to address **C1**, we introduce Federated Matrix Selection (FMS) as our first module, which selectively aggregates the $B$ matrices across clients while keeping the $A$ matrices local, achieving efficient collaboration without exposing private representations. To address **C2**, we expect to isolate each client's personalized updates, ensuring that their distinct knowledge remains protected after distribution. Therefore, we introduce Personalized Matrix-level Adaptation (PMA), which constructs a decorrelated projection matrix that encourages each client's $B$ updates to be nearly independent to the $A$ matrices of other clients. This projection defines a subspace that is maximally unaligned across clients, thereby reducing cross-client interference and enhancing personalized adaptation.

To validate our approach, we conduct comprehensive experiments across multiple datasets using the Roberta and ViT architectures, demonstrating the superior performance in both natural language and vision classification aspects. Our contributions are summarized as follows:

- We reveal key issues in the current pFedLoRA approaches due to its reliance on a dual-LoRA architecture, *i.e.*, excessive computational overhead and structural redundancy. To mitigate these issues, we then propose to explore single-LoRA approaches to achieve personalization by leveraging its intrinsic structural characteristics.
- We propose FedPissa, a novel framework that leverages selective $B$ matrix sharing and decorrelated subspace construction to separate shared and private representations within a single-LoRA architecture.
- Extensive experiments on the GLUE using RoBERTa-Large and on the DomainNet using ViT-Base validate the effectiveness of FedPissa under both feature non-IID and feature-and-label non-IID settings. FedPissa achieves up to 8% higher accuracy and 35% lower computation cost compared with existing pFedLoRA baselines.

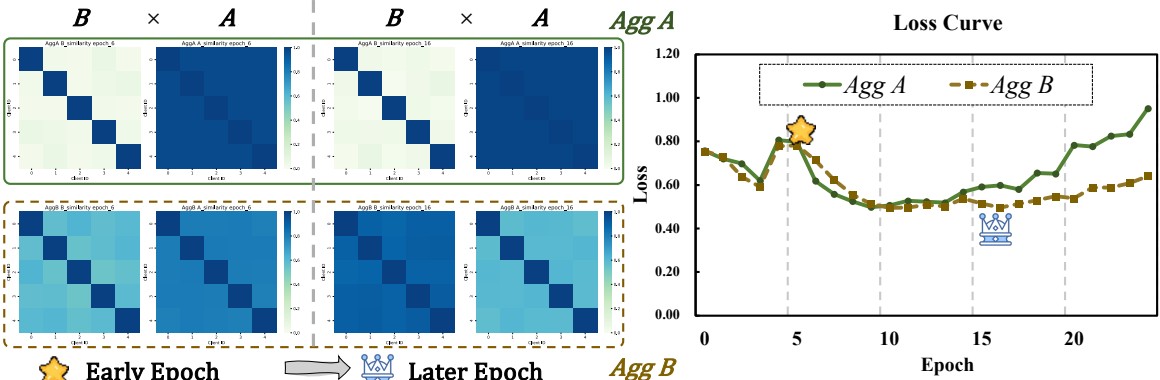

*Figure 2.* Visualization of client similarity and convergence trends under different aggregation methods. Heatmaps show inter-client similarities of $A$ and $B$ at early and later epochs, while the line chart depicts the corresponding loss curves for Agg $A$ and Agg $B$. Agg $A$ brings nearly no change for $A$ and $B$ and Agg $B$ brings slowly personalization for $A$ and faster converge for $B$.

## 2. Related Work

### 2.1. Federated Foundation Tuning

Fine-tuning LLMs has become a standard approach for adapting general-purpose pre-trained models to specific downstream tasks (Yang et al., 2021; OpenAI, 2023; Lu et al., 2024a; Qi et al., 2026). However, traditional centralized fine-tuning requires aggregating sensitive user data on a central server, which raises growing privacy and regulatory concerns. To address these limitations, federated fine-tuning (Yu et al., 2025; Yi et al., 2025) has emerged as a promising paradigm that enables distributed clients to fine-tune LLMs locally on private datasets while sharing only model updates with a coordinating server. This privacy-preserving design aligns with modern data protection requirements and user expectations. Nevertheless, the massive scale of LLM parameters and the inherent heterogeneity of client data introduce substantial technical challenges. One of the most important keys includes high communication overhead. To overcome the challenge, researchers have proposed various parameter-efficient federated fine-tuning methods (Mangrulkar et al., 2022; Zhang et al., 2025; 2023; Zhuang et al., 2023) that strategically reduce resource consumption while maintaining downstream task performance.

### 2.2. Low-Rank Adaptation (LoRA)

Parameter-Efficient Fine-Tuning (PEFT) methods adapt large language models (LLMs) to downstream tasks with greatly reduced computation and storage costs (Mangrulkar et al., 2022). A widely adopted technique in PEFT is Low-Rank Adaption (LoRA) (Hu et al., 2022). LoRA decomposes the weight update into two trainable low-rank matrices added in parallel to the frozen pre-trained weights. Only these low-rank parameters are updated during training, and they can be merged back at inference. Existing solutions can be primarily categorized into two types. **i)** *Inherent Calibration Reparameterization.* It refines the internal optimization of LoRA via parameter regularization, orthogonal constraints, or adaptive learning rates. These adjustments enhance training stability and generalization (Kopiczko et al., 2023; Wu et al., 2024; Agiza A, 2024). **ii)** *External Structure Reparameterization.* It restructures the LoRA architecture with modular designs or expert-based combinations. Such structural reparameterization improves representation capacity and task adaptability (Hayou et al., 2024; Lu et al., 2024b; Wang et al., 2024). While LoRA improves efficiency on single-domain tasks, it assumes centralized data. In practice, data are distributed and private, motivating federated variants for collaborative tuning without raw data sharing.

### 2.3. Federated Learning Meets LoRA

Building on the efficiency of LoRA, FedLoRA integrates PEFT into federated learning, enabling collaborative training without sharing raw data. FedIT (Zhang et al., 2024a) pioneers this direction but lacks personalization under data heterogeneity (Ye et al., 2023; Huang et al., 2022; 2024). To address this, Personalized FedLoRA (pFedLoRA) (Yi et al., 2023; Fang & Ye, 2022) enhances adaptability across diverse clients and has evolved into two paradigms: 1) Static coordination (Guo et al., 2025b; Sun et al., 2024; Yang et al., 2024) methods like FedSA-LoRA share $A$ globally while keeping $B$ local; FFA-LoRA fine-tunes zero-initialized $B$ for task-specific adaptation; FedDPA uses dual adapters to alternate between global and local updates. In contrast, 2) Dynamic clustering (Wang et al., 2025; Bian et al., 2025; Zhang et al., 2024b) approaches such as FedLEASE employ routing mechanisms to group similar clients and assign cluster-specific LoRA experts. Most pFedLoRA methods rely on a dual-LoRA architecture, maintaining one module for global knowledge and another for local adaptation—introducing redundancy and computational overhead. This motivates our work: a more efficient single-LoRA approach for scalable personalized federated learning.

## 3. Preliminary

**LoRA** (Hu et al., 2022) is a widely adopted PEFT technique that achieves performance comparable to full fine-tuning by introducing trainable low-rank matrices into pre-trained model weights. Given a pre-trained linear layer with weight parameters $\mathbf{W}_0 \in \mathbb{R}^{l \times d}$, where $d$ and $l$ denote the input and output dimensions, respectively, LoRA decomposes the weight update into two sequential low-rank matrices $\boldsymbol{A} \in \mathbb{R}^{r \times d}$ and $\boldsymbol{B} \in \mathbb{R}^{l \times r}$, where $r \ll \min(d, l)$. The adapted output can be formulated as:

$$\boldsymbol{y} = \mathbf{W}_0 \boldsymbol{x} + \boldsymbol{B}\boldsymbol{A}\boldsymbol{x}, \tag{1}$$

where $\boldsymbol{A}$ is initialized with random Gaussian values and $\boldsymbol{B}$ is initialized with zeros to ensure stable optimization in the early stages of fine-tuning. By keeping $\mathbf{W}_0$ frozen and updating only $\boldsymbol{A}$ and $\boldsymbol{B}$, LoRA significantly reduces the number of trainable parameters and memory footprint while preserving model performance across downstream tasks.

**Federated LoRA** (McMahan et al., 2017) enables multiple clients to collaboratively train a model under the coordination of a central server without sharing private data. In each communication round $t$, the server broadcasts the global model $\mathbf{W}_g^t$ to selected clients, which locally update it on their private datasets $\mathcal{D}_k$ and return the adapted parameters $\mathbf{W}_k^{t+1}$. The server then aggregates the received updates by $\mathbf{W}_g^{t+1} = \sum_{k=1}^{K} \frac{N_k}{N} \mathbf{W}_k^{t+1}$, where $N_k$ denotes the local data size, *i.e.*, $N_k = |D_k|$. The $N$ denotes the overall data scale across clients, *i.e.*, $N = \sum_k N_k$. When combined with LoRA (Zhang et al., 2024a), each client fine-tunes only the low-rank matrices $\boldsymbol{A}_k$ and $\boldsymbol{B}_k$ while keeping the pre-trained weights $\mathbf{W}_0$ frozen, and the server performs aggregation on these low-rank parameters as the following formualtions:

$$\{\boldsymbol{A}^{t+1}, \boldsymbol{B}^{t+1}\} = \sum_{k=1}^{K} \frac{N_k}{N} \{\boldsymbol{A}_k^{t+1}, \boldsymbol{B}_k^{t+1}\}. \tag{2}$$

This pipeline (Zhang et al., 2024a) effectively reduces communication and computation costs during federated fine-tuning, but also introduces conflict in personalization.

## 4. Proposed Method

### 4.1. Motivation

The dual-LoRA architecture used in existing pFedLoRA frameworks incurs high computational cost and structural redundancy, as each client must maintain and activate two LoRA modules during training and inference. Specifically, the output of a dual-LoRA model can be formulated as:

$$\boldsymbol{y} = \boldsymbol{W}_0 \boldsymbol{x} + (\boldsymbol{B}^g \boldsymbol{A}^g + \boldsymbol{B}^l \boldsymbol{A}^l)\boldsymbol{x}, \tag{3}$$

where $(\boldsymbol{A}^g, \boldsymbol{B}^g)$ and $(\boldsymbol{A}^l, \boldsymbol{B}^l)$ denote the global and local LoRA modules, respectively. This structure doubles the forward computation compared to a single-LoRA setting, leading to increased latency and memory usage. These findings highlight the inefficiency of maintaining dual modules for personalization, which naturally raises the challenge C1.

Although identifying the appropriate component for sharing can alleviate the computational burden, direct aggregation across heterogeneous clients may still lead to personalized knowledge conflicts due to the non-IID nature of local data. When clients possess diverse data distributions, their locally optimized LoRA parameters often converge toward different optima in the parameter space. Averaging these updates in a global aggregation step can blur task-specific features and even distort the shared representation, ultimately degrading personalization performance. This inherent conflict between global consistency and local specialization naturally raises the challenge C2 (see Sec. 1).

### 4.2. Federated Matrix Selection

To address **C1** (the dilemma of selecting which LoRA component to personalize), we propose FMS, a selective federated fine-tuning strategy that identifies the most suitable LoRA component for global aggregation. We conduct an observational study comparing evaluation loss under two aggregation schemes: aggregating only $\boldsymbol{A}$ (*Agg* $\boldsymbol{A}$) (Guo et al., 2025b) or only $\boldsymbol{B}$ (*Agg* $\boldsymbol{B}$)(shown in Figure 2). When only $\boldsymbol{A}$ is aggregated, inter-client similarity remains nearly static throughout training. Since all clients start with similar $\boldsymbol{A}$ matrices and update $\boldsymbol{B}$ locally, client models diverge gradually due to data heterogeneity and always aggregate static $\boldsymbol{A}$ matrix, leading to poor coordination and unstable convergence.

In contrast, aggregating $\boldsymbol{B}$ (*Agg* $\boldsymbol{B}$) significantly reduces model divergence by aligning the fast-adapting, high-variance components across clients. Meanwhile, the locally retained $\boldsymbol{A}$ matrices evolve slowly, preserving client-specific characteristics and enabling personalized adaptation. This leads to faster and more stable convergence, as shown by the lower and smoother loss curve of *Agg* $\boldsymbol{B}$ compared to *Agg* $\boldsymbol{A}$ in Figure 2. Furthermore, inter-client similarity analysis reveals that $\boldsymbol{A}$ maintains high consistency across clients, while $\boldsymbol{B}$ exhibits substantial variation, reflecting divergent local optimization paths and rates. These observations indicate that $\boldsymbol{A}$ evolves homogeneously and thus is better suited for personalization, whereas $\boldsymbol{B}$ captures shared directional updates despite its initial heterogeneity. Therefore, aggregating $\boldsymbol{B}$ enhances global coordination and improves local adaptation, supporting our key design: leveraging $\boldsymbol{B}$ for shared knowledge and $\boldsymbol{A}$ for client-specific learning within a single LoRA module.

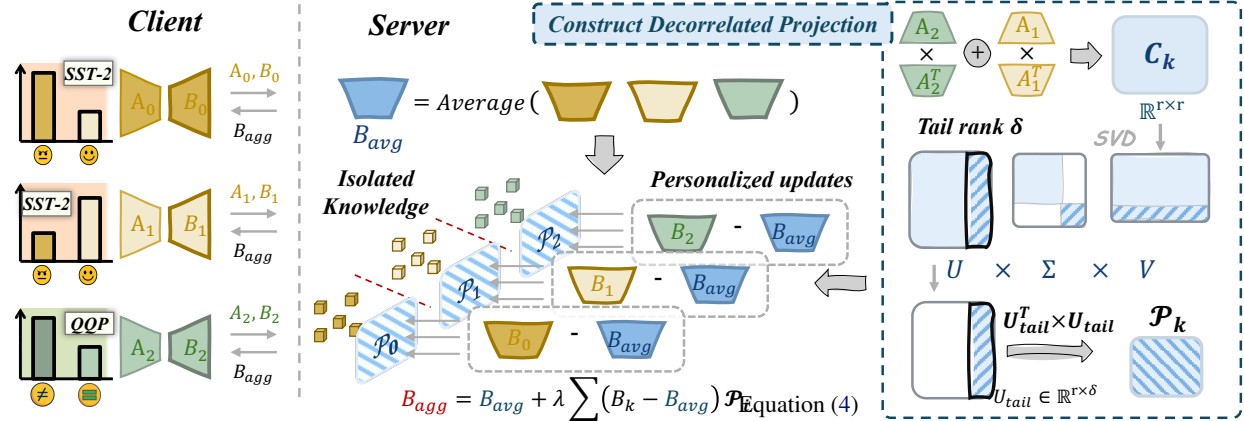

*Figure 3.* Schematization of FedPissa. We selectively aggregates the $B$ matrices from clients ( ◣, ◻, ◣ ) and constructs a decorrelated subspace projection matrix ( ◪ ) on the server to refine the aggregated updates. More details can be found in Section 4.

### 4.3. Personalized Matrix-level Adaptation

Addressing **C2** remains challenging in the single-LoRA paradigm. While dual-LoRA frameworks use expert routing (*e.g.*, MoE) to reduce cross-client interference, such mechanisms are infeasible in a compact single-LoRA setting, where simple averaging of $B$ often causes *personalization conflicts*. We first show the entire action, and then we specifically introduce the decorrelated projection matrix. The relative proof can be found in Section C.1.

**Federated Decorrelation Aggregation.** To mitigate the personalization conflicts that arise from directly averaging heterogeneous $B_k$ matrices, We propose a decorrelated aggregation strategy formulated as follows:

$$B_{\text{agg}} = B_{\text{avg}} + \lambda \sum_{i=1}^{K} (B_k - B_{\text{avg}}) \mathcal{P}_k, \qquad (4)$$

where $B_{\text{avg}} = \sum_k p_k B_k$ denotes the standard FedAvg aggregation result and $\lambda$ controls the strength of the decorrelation compensation. The residual term $(B_k - B_{\text{avg}})$ captures the personalized deviation that is typically lost in simple averaging, while the projection matrix $\mathcal{P}$ reintroduces this information in a controlled and decorrelated manner.

**Construction of the Decorrelation Matrix.** To construct $\mathcal{P}_k$, each client first computes other clients local covariance matrix $A_i A_i^\top$, and the server then aggregates them into a global covariance structure with simple average:

$$\mathbf{C}_k = \sum_{i=1, k \neq i} \alpha_i A_i A_i^\top, \qquad (5)$$

where $\alpha_i = \frac{N_i}{N - N_k}$. The server then extracts tail subspace spanned by the least significant singular vectors of $\mathbf{C}$:

$$\mathcal{P}_k = \text{TailRank}_\delta(\mathbf{C}_k), \qquad (6)$$

where $\text{TailRank}_\delta(\cdot)$ denotes selecting the $r$ **smallest** eigencomponents (rank r to the end). This process effectively sup-

presses dominant shared directions while preserving complementary and decorrelated bases for personalized updates. In doing so, it approximates the objective:

$$\min_{k \neq i} \|(B_k - B_{\text{avg}}) A_i\|_F^2, \qquad (7)$$

thereby reducing representational coupling across heterogeneous clients. The proof can be found in Section C.1.

**Client Update.** After obtaining $B_{\text{agg}}$, the server distributes it to all clients for next round. For each client $k$, the decorrelation projection $\mathcal{P}_k$ ensures that the residual term $(B_k - B_{\text{avg}}) \mathcal{P}_k$ selectively enhances its own subspace while remaining nearly decorrelated to those of other clients. Thus, when client $k$ receives $B_{\text{agg}}$, the projected residual helps recover its lost personalization, whereas for other clients, the same term is largely neutralized due to subspace decorrelation. This mechanism allows each client to benefit from shared knowledge via $B_{\text{avg}}$ while maintaining local adaptation through its personalized residual module.

### 4.4. Discussion and Limitation

**Privacy Concern**. According to (Guo et al., 2025b), the closed form for each client is $B_k^\star = \Delta W_k \mathbb{E}[xx^\top] Q^\top (Q \mathbb{E}[xx^\top] Q^\top)^{-1}$, where $Q \equiv A$ and $\mathbb{E}[xx^\top]$ denotes the local distribution. If $A$ is not distributed to other clients, an adversary who only observes $B_k^\star$ cannot recover the local covariance $\mathbb{E}[xx^\top]$: it only appears in the composite term $\mathbb{E}[xx^\top] A^\top (A \mathbb{E}[xx^\top] A^\top)^{-1}$, which is non-identifiable without $A$. Thus, uploading and aggregating $B$ alone does not reveal client data distribution statistics; the mapping from $\mathbb{E}[xx^\top]$ to $B_k^\star$ is many-to-one in the absence of $A$.

**Limitation.** Although our method shows promise in pFed-LoRA, it has several limitations. First, our approach relies on tail rank projection, which may not be effective for extremely small LoRA ranks. In such cases, the tail and top

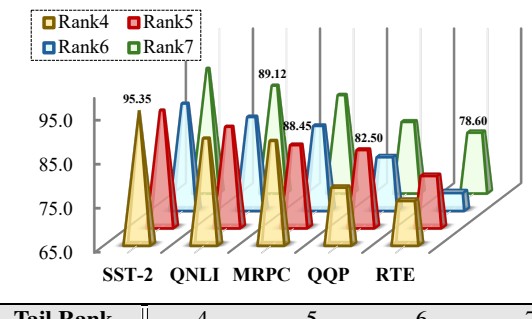

| Tail Rank | 4 | 5 | 6 | 7 |
|---|---|---|---|---|
| Mean | 83.35 | 83.88 | 80.33 | 84.33 |

*Figure 4.* Abaltion perforamnce under different tail subspace ranks under feature non-IID. Rank $x$ denotes $x$ to the end of LoRA rank. While Rank7 yields the highest mean, Rank5 achieves more stable performance. More details can be found in Section 5.2.

*Table 1.* Ablation performance on different $\lambda$ values under the feature non-IID setting. Arrows indicate absolute performance changes relative to FedIT. The detailed explanation and selected configurations are provided in Section 5.2.

| $\lambda$ | GLUE | DomainNet |
|---|---|---|
| 0.05 | 82.48 $_{\uparrow 10.51}$ | **51.80** $_{\uparrow 8.07}$ |
| 0.1 | **83.88** $_{\uparrow 11.91}$ | 50.55 $_{\uparrow 6.82}$ |
| 0.5 | 76.85 $_{\uparrow 4.88}$ | 49.48 $_{\uparrow 5.75}$ |
| 1.0 | 81.98 $_{\uparrow 10.01}$ | 49.98 $_{\uparrow 6.25}$ |

*Table 2.* Ablation of different aggregation strategies on six subdomains of DomainNet under feature non-IID. Columns use abbreviations for notations: Q (Quickdraw), C (Clipart), P (Painting), S (Sketch), R (Real), I (Infograph). See details in Section 5.2.

| FMS | PMA | Q | C | P | S | R | I | Mean |
|---|---|---|---|---|---|---|---|---|
| ✗ | ✗ | 54.20 | 34.70 | 43.80 | 29.20 | 53.60 | 46.90 | 43.73 |
| ✓ | ✗ | 60.30 | **38.20** | 48.50 | 38.30 | 70.90 | 53.90 | 51.68 |
| ✓ | ✓ | **61.40** | 36.10 | **48.50** | **39.20** | **71.30** | **54.30** | **51.80** |

ranks may interfere due to excessive compression, making it difficult to extract a well-defined, decorrelated subspace. Consequently, performance may degrade, consistent with the broader challenge that highly compressed representations inherently limit expressiveness.

Second, under homogeneous data distributions, where clients have similar data, the tail rank projection may hinder the capture of shared generalization patterns, leading to a slight performance drop. However, in practice, perfect homogeneity is rare; even subtle differences exist between clients. As the adage goes: *"No two snowflakes are alike."* Thus, our method FedPissa remains robust and effective in real-world scenarios characterized by diverse and various heterogeneous data distributions setting.

*Table 3.* Effect of local training epochs under the feature non-IID setting. We report the mean accuracy over SST-2, QNLI, and MRPC. The best result under each local-epoch setting is highlighted in bold. More details can be found in Section 5.2.

| Local Epochs | Methods | SST-2 | QNLI | MRPC | Mean |
|---|---|---|---|---|---|
| 2 | FedIT | **93.10** | 76.50 | 73.10 | 80.90 |
| | FedLEASE | 92.10 | 80.50 | 71.00 | 81.20 |
| | **Ours** | 91.40 | **87.70** | **83.60** | **87.57** |
| 3 | FedIT | 91.00 | 79.80 | 70.60 | 80.47 |
| | FedLEASE | 91.10 | **89.00** | 77.50 | 85.87 |
| | **Ours** | **93.60** | 84.40 | **82.40** | **86.80** |
| 4 | FedIT | 93.20 | 83.50 | 75.70 | 84.13 |
| | FedLEASE | 92.60 | 80.50 | 79.30 | 84.13 |
| | **Ours** | **96.50** | **88.50** | **88.00** | **91.00** |
| 5 | FedIT | 92.00 | 83.20 | 69.80 | 81.67 |
| | FedLEASE | 89.50 | 75.10 | 76.70 | 80.43 |
| | **Ours** | **94.20** | **85.90** | **87.60** | **89.23** |

*Table 4.* Effect of global communication rounds under the feature non-IID setting. We report the mean accuracy over SST-2, QNLI, and MRPC. The results with 25 global rounds are taken from Table 5. The best result under each global-round setting is highlighted in bold. More details can be found in Section 5.2.

| Global Rounds | Methods | SST-2 | QNLI | MRPC | Mean |
|---|---|---|---|---|---|
| 25 | FedIT | **93.10** | 76.50 | 73.10 | 80.90 |
| | FedLEASE | 92.10 | 80.50 | 71.00 | 81.20 |
| | **Ours** | 91.40 | **87.70** | **83.60** | **87.57** |
| 30 | FedIT | 90.40 | 75.20 | 72.80 | 79.47 |
| | FedLEASE | 91.80 | 80.60 | 77.10 | 83.17 |
| | **Ours** | **94.80** | **85.30** | **85.40** | **88.50** |
| 35 | FedIT | **94.90** | 78.50 | 72.10 | 81.83 |
| | FedLEASE | 92.00 | **88.10** | 83.10 | 87.73 |
| | **Ours** | 94.10 | 87.50 | **86.00** | **89.20** |

# 5. Experiments

## 5.1. Experimental Setup

**Foundation Models.** We employ two representative foundation models from text to vision to evaluate the effectiveness and personalized performance of our approach:

- **RoBERTa-large** (Liu et al., 2019): a transformer-based language model pre-trained on large textual corpora. It is our backbone for natural language understanding tasks.

- **ViT-B/16** (Dosovitskiy et al., 2021): a vision transformer pre-trained on the ImageNet-21k dataset for classification.

These two models provide a unified evaluation framework for examining both linguistic and visual classification under federated personalized adaptation settings.

**Datasets.** We evaluate our method on two modalities of benchmark datasets (GLUE and DomainNet) to comprehensively assess its capabilities. More details see in Table 7:

- **GLUE** (Wang et al., 2018): For natural language understanding, we use six representative datasets from GLUE: SST-2, QNLI, MRPC, QQP, RTE, and MNLI. These datasets cover diverse linguistic tasks such as sentiment analysis, language inference, and paraphrase detection.

*Table 5.* Performance comparison across multiple GLUE tasks between single-LoRA and dual-LoRA methods. We report the mean accuracy over the last 5 epochs. For FedPissa, arrows indicate absolute accuracy changes relative to FedIT under the same non-IID setting.

| LoRA Type | Methods | Performance Accuracy↑ | | | | | | |
|---|---|---|---|---|---|---|---|---|
| | | SST-2 | QNLI | MRPC | QQP | RTE | MNLI | Mean |
| *Feature non-IID* | | | | | | | | |
| Dual LoRA | FedDPA | 92.80 | 83.30 | 78.80 | 78.70 | 71.60 | 62.70 | 77.98 |
| | FedLEASE | 92.10 | 80.50 | 71.00 | 72.30 | 54.20 | 72.30 | 73.73 |
| Single LoRA | FedIT | 93.10 | 76.50 | 73.10 | 69.70 | 58.30 | 61.10 | 71.97 |
| | FedSA | 89.70 | 81.20 | **85.00** | 81.00 | 76.60 | 78.80 | 82.05 |
| | FFA-LoRA | **94.70** | 83.30 | 80.10 | 81.60 | 68.80 | 44.20 | 75.45 |
| | FedPissa | 91.40 ↓1.70 | **87.70** ↑11.20 | 83.60 ↑10.50 | **82.50** ↑12.80 | **76.70** ↑18.40 | **81.40** ↑20.30 | **83.88** ↑11.91 |
| *Feature and Label non-IID ($\beta = 0.5$)* | | | | | | | | |
| Dual LoRA | FedDPA | **93.50** | 77.80 | 78.45 | 81.75 | 64.50 | 73.70 | 78.28 |
| | FedLEASE | 91.85 | 82.50 | 81.55 | 76.75 | 55.30 | 57.45 | 74.23 |
| Single LoRA | FedIT | 88.65 | 76.80 | 72.85 | 74.85 | 60.60 | 47.70 | 70.24 |
| | FedSA | 92.60 | 80.40 | **84.15** | 83.40 | 63.00 | **77.15** | 80.12 |
| | FFA-LoRA | 92.35 | 76.65 | 72.35 | 83.15 | **66.50** | 58.70 | 74.95 |
| | FedPissa | 92.20 ↑3.55 | **83.75** ↑6.95 | 83.70 ↑10.85 | **85.00** ↑10.15 | 66.45 ↑5.85 | 76.45 ↑28.75 | **81.26** ↑11.02 |
| *Feature and Label non-IID ($\beta = 0.1$)* | | | | | | | | |
| Dual LoRA | FedDPA | 98.25 | **86.55** | 75.40 | 85.45 | 84.85 | **87.25** | 86.29 |
| | FedLEASE | 98.00 | 72.60 | 63.35 | 85.20 | 86.90 | 81.10 | 81.19 |
| Single LoRA | FedIT | 53.95 | 69.30 | 69.70 | 70.40 | 58.35 | 53.05 | 62.46 |
| | FedSA | 98.10 | 86.40 | 84.50 | 85.40 | 89.20 | 86.15 | 88.29 |
| | FFA-LoRA | 71.25 | 83.35 | 69.90 | 78.95 | 82.60 | 71.95 | 76.33 |
| | **FedPissa** | **98.70** ↑44.75 | 86.05 ↑16.75 | 84.50 ↑14.80 | **88.40** ↑18.00 | **89.65** ↑31.30 | 87.25 ↑34.20 | **89.09** ↑26.63 |

*(a)* Feature non-IID      *(b)* Feature and Label non-IID ($\beta = 0.5$)      *(c)* Feature and Label non-IID ($\beta = 0.1$)

*Figure 5.* Mean accuracy comparison during federated process with counterparts under various heterogeneous degrees. See Section 5.3.

- **DomainNet** (Peng et al., 2019): For visual classification, we adopt domains: Quickdraw, Clipart, Painting, Sketch, Real, and Infograph. They exhibit substantial distributional discrepancies across visual styles and content categories, providing a challenging benchmark for evaluating cross-domain adaptation.

**Heterogeneity Setting.** To simulate heterogeneous data distributions across clients, we consider two scenarios:

- **Domain Shift** (Volpi & Murino, 2019): Client data are partitioned with distinct feature distributions while sharing the same label space, capturing domain-specific variations such as diverse writing styles or visual textures.

- **Domain and Label Shift** (Wang et al., 2025): We introduce label imbalance via Dirichlet distribution ($\beta$), modeling heterogeneity in both feature and label spaces.

**Training Details.** All experiments are conducted under consistent hyperparameter settings for fair comparison. The training batch size is set to 128, and the evaluation batch size to 256. Each client performs 2 local iterations per communication round. We train for 25 rounds on the GLUE benchmark and 50 rounds on the DomainNet dataset. The learning rate is fixed at $1 \times 10^{-3}$, and the optimizer is AdamW. In the *Feature non-IID* setting, we simulate 6 clients, while in the *Feature and Label non-IID* setting, the number of clients

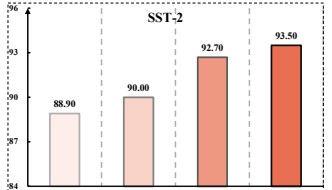 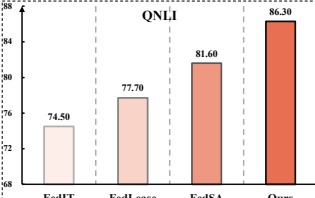 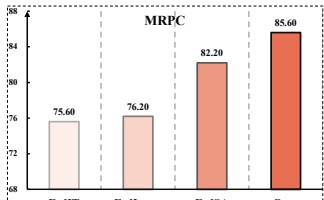 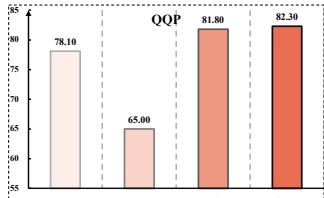

*Figure 6.* Ablation results with a larger LoRA rank ($r = 16$) under the feature non-IID setting. FedPissa still achieves the best performance across all evaluated GLUE tasks when the LoRA rank is increased to 16. More details can be found in Section 5.2.

increases to 12. Moreover, We compute the mean and var of the last 5 epochs for the final result.

**Counterparts.** We compare against representative pFed-LoRA methods, categorized by their architectural design.

First, for the Single-LoRA formulation:
- FedIT [ICASSP'24] (Zhang et al., 2024a) directly integrates LoRA into federation, enabling collaborative fine-tuning without sharing raw data.
- FFA-LoRA [ICLR'24] (Sun et al., 2024) fixes the randomly initialized non-zero $A$ matrices and fine-tunes only the zero-initialized $B$ matrices the across training rounds. And then, FFA aggregates $B$ matrices in the server.
- FedSA [ICLR'25] (Guo et al., 2025b) shares $A$ matrices capturing general knowledge while keeping personalized $B$ matrices local.

Second, with respect to the Dual-LoRA formulation:
- FedDPA [NeurIPS'24] (Yang et al., 2024) propoes a dual-adapter structure to jointly address distribution and client heterogeneity.
- FedLEASE [NeurIPS'25] (Wang et al., 2025) employs an adaptive mixture-of-experts mechanism to clusters clients.

### 5.2. Diagnostic Analysis

**Subspace Rank Selection.** As shown in Fig. 4, although higher ranks yield strong performance on simpler tasks such as SST-2, QNLI, and MRPC, they tend to overfit and lose effectiveness on semantic understanding tasks like QQP and RTE, which require more structured subspace separation. Therefore, we select **rank = 5** as the default value for all subsequent experiments, with competitive performance.

**Lambda $\lambda$ Selection.** As shown in Table 1, the performance varies with the decorrelation strength $\lambda$. For language understanding tasks on **GLUE**, $\lambda = \mathbf{0.1}$ achieves the most consistent results across tasks, balancing generalization and personalization. For visual classification on **DomainNet**, a smaller $\lambda = \mathbf{0.05}$ yields better performance, indicating that mild decorrelation better preserves domain-specific visual representations. Therefore, we set $\lambda = 0.1$ for language tasks and $\lambda = 0.05$ for vision tasks in all subsequent experiments. More details see in Figure 9.

**Target Objective.** Table 2 compares three aggregation strategies across six subdomains of DomainNet under the feature non-IID setting. The configuration (✗, ✗) corresponds to our baseline FedIT, which aggregates both $A$ and $B$ matrices, leading to insufficient domain adaptation. When only $B$ is aggregated *i.e.*, (✗, ✓), the performance notably improves, indicating that $B$ carries richer personalized information. Finally, our proposed configuration (✓, ✓) further enhances performance by incorporating decorrelated mapping, effectively mitigating cross-client interference and achieving a better personalized specialization. More details see in Figure 10.

**Training Schedules.** We study the effect of local epochs and global communication rounds under the feature non-IID setting. As shown in Tables 3 and 4, FedPissa consistently achieves the best mean accuracy across different settings, showing stable performance under varying training schedules. Considering both efficiency and performance, we use 2 local epochs and 25 global communication rounds as the default setting in our main experiments.

**LoRA Rank.** We further evaluate the performance with a larger LoRA rank, *i.e.*, $r = 16$, while using $r = 8$ as the default setting in our main experiments. As shown in Figure 6, FedPissa consistently outperforms the compared methods on all evaluated tasks, indicating that our method remains effective when the LoRA adaptation capacity increases.

### 5.3. Comparison to State-of-the-Arts

**Natural Language Understanding.** Table 5 compares our method with representative pFedLoRA baselines on the GLUE benchmark under Feature non-IID and Feature and Label non-IID settings. In the Feature non-IID setting, Fed-Pissa achieves the best average accuracy (**83.88**), with clear gains on QNLI, QQP, RTE, and MNLI. Under Feature and Label non-IID with $\beta = 0.5$, FedPissa also obtains the highest mean accuracy (**81.26**), outperforming both single-LoRA and dual-LoRA baselines. When the label heterogeneity becomes stronger with $\beta = 0.1$, FedPissa further achieves the best overall performance (**89.09**) and remains competitive across all tasks. These results show that selective aggregation and decorrelated subspace learning improve robustness and adaptation under heterogeneous data distributions.

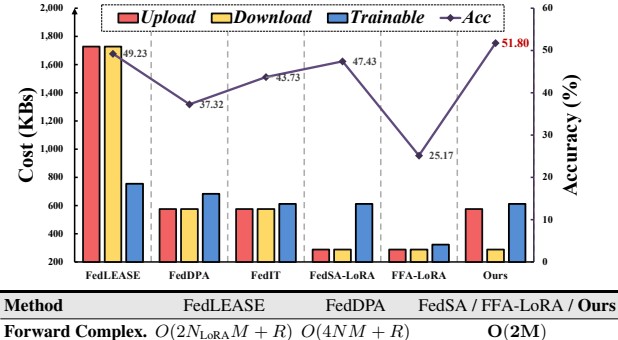

| Method | FedLEASE | FedDPA | FedSA / FFA-LoRA / **Ours** |
|---|---|---|---|
| **Forward Complex.** | $O(2N_{\text{LoRA}}M + R)$ | $O(4NM + R)$ | **$O(2M)$** |

*Figure 7.* Communication and computation cost analysis. The upper figure compares upload/download/trainable parameters and accuracy. The lower table summarizes forward-propagation complexity. $M$ denotes the half-LoRA structure, $R$ represents router memory, and $N_{\text{LoRA}}$ the number of LoRA modules.

**Vision Classification.** Table 8 reports the results across six subdomains of DomainNet under the feature non-IID setting. Dual-LoRA methods such as FedDPA and FedLEASE demonstrate moderate performance, but their dual-module design incurs higher computation cost and limited scalability. Among Single-LoRA baselines, FedAvg struggles to generalize under domain variation, while FedSA and FFA-LoRA show inconsistent performance due to partial parameter sharing or fixed low-rank adaptation. In contrast, our method achieves the highest average accuracy (**51.80**), with consistent gains across most subdomains, particularly in `Real` and `Infograph`. These results validate the effectiveness of selectively aggregating $B$ and decorrelating client updates, which enables our approach to maintain robustness and personalized adaptation across diverse visual domains and visual tasks.

**Training Process Curve.** To further analyze the optimization behavior, we visualize the convergence trajectories on both GLUE and DomainNet in Fig. 5. Compared to existing single-LoRA methods such as FedAvg and FedSA-LoRA, our method exhibits consistently faster convergence and *more stable* performance improvement across rounds. On GLUE, selectively aggregating $B$ enables more effective propagation to capture adaptive knowledge, leading to rapid gains and stabilizing after around 15 communication rounds. On DomainNet, FedPissa converges faster while remaining robust under highly heterogeneous visual domains, further confirming the efficiency and stability of our method.

**Computation and Efficiency Analysis.** Fig. 7 presents the comparison of communication cost, trainable parameters, and forward complexity. Dual-LoRA methods such as FedLEASE and FedDPA rely on stacking multiple LoRA experts to improve adaptability, which results in substantially higher computational and communication overhead. Although FedLEASE achieves competitive accuracy, its design requires $O(2N_{\text{LoRA}}M + R)$ complexity due to multi-expert routing. In contrast, FFA-LoRA and FedSA min-

imize computation to $O(2M)$ but suffer from severe performance degradation. Our method maintains a compact single-LoRA structure, reaching comparable or superior accuracy with minimal computational cost, effectively balancing performance and efficiency.

## 6. Conclusion

We revisit the limitations of existing dual-LoRA architectures for pFedLoRA, which incur substantial computational overhead and structural redundancy. While the single-LoRA paradigm offers a more compact design, it poses new challenges for knowledge coupling and conflict. To address this, we propose **FedPissa**, the first framework that rethinks single-LoRA from the perspective of selective aggregation and subspace decorrelation. Specifically, FedPissa selectively aggregates the adaptive $B$ component and projects personalized updates into decorrelated subspaces to mitigate cross-client interference. Extensive experiments across textual and visual benchmarks demonstrate that FedPissa achieves effective knowledge integration and better personalization with reduced computation and communication costs.

## Acknowledgement

This work is funded by National Key Research Development Program of China ( 2026YFE0202100), National Natural Science Foundation of China under Grant (62361166629, 623B2080), the Major Project of Science and Technology Innovation of Hubei Province (2024BCA003), Key Research and Development Program of Wuhan (2025061202030423), the Innovative Research Group Project of Hubei Province under Grants 2024AFA017, and the NTU AI-for-X Postdoctoral Fellowship. The supercomputing system at the Supercomputing Center of Wuhan University supported the numerical calculations in this paper.

## Impact Statement

This paper presents work whose goal is to advance the field of machine learning. There are many potential societal consequences of our work, none of which we feel must be specifically highlighted here.

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

**Algorithm 1 FedPissa**

---

**Data:** Frozen pre-trained weights $W_0$; learning rate $\eta$; decorrelation strength $\lambda$; number of clients $K$; local epochs $E$; total rounds $T$.

**Input:** LoRA matrices $\{A_k, B_k\}_{k=1}^K$ from each client.

**Output:** Personalized $\{A_k, B_k\}$ for each clients.

**for** $t = 1, 2, \ldots, T$ **do**

   *Participant Side:*    Each client $k$ receives the aggregated $B_{\mathrm{agg}}^t$   Update local matrices for $E$ epochs   Upload updated $(A_k^{t+1}, B_k^{t+1})$ to the server

   ***Server Side:***      `// --- Federated Matrix Selection (FMarS) ---`

   Compute weighted mean of $B$:

$$B_{\mathrm{avg}} = \sum_{k=1}^K p_k B_k^{t+1}, \quad p_k = \frac{N_k}{N}$$

   `// --- Personalized Matrix-level Adaptation (PerMA) ---`

   For each client $k$, construct covariance:

$$C_k = \sum_{i \neq k} p_i A_i A_i^\top$$

   Obtain decorrelated projection matrix:

$$P_k = \mathrm{TailRank}_r(C_k)$$

   Compute decorrelated aggregation:

$$B_{\mathrm{agg}}^{t+1} = B_{\mathrm{avg}} + \lambda \sum_{k=1}^K p_k (B_k^{t+1} - B_{\mathrm{avg}}) P_k$$

**end**   Distribute $B_{\mathrm{agg}}^{t+1}$ to all clients.

**Return:** $\{A_k, B_k\}_{k=1}^K$

---

## A. Notation Table

We provide the notation table in Table 6.

*Table 6.* **Notations** table.

| Symbol | Description | Link |
|---|---|---|
| $A_k$ | Matrix $A$ for client $k$ | Equation (2) |
| $B_k$ | Matrix $B$ for client $k$ | Equation (2) |
| $C_k$ | Representation for all clients except client $k$ | Equation (6) |
| $P_k$ | Decorrelated projection matrix for client $k$ | Equation (5) |
| $\alpha_k$ | Aggregated weight for client $k$ | Equation (5) |
| $N_k$ | Number of data samples for client $k$ | Equation (2) |
| $N$ | Total number of data samples in the entire FL system | Equation (2) |
| $A_{\mathrm{avg}}$ | Simple aggregation of $A$ across clients | - |
| $B_{\mathrm{avg}}$ | Simple aggregation of $B$ across clients | Equation (4) |
| $B_{\mathrm{agg}}$ | FedPissa aggregation of $B$ distributed to all clients | Equation (4) |

## B. Algorithm of FedPissa

We provide the algorithm of our method in Algorithm 1.

## C. Theoretical Analysis

The following subsection provides the theoretical justification that the tail-eigenspace of the aggregated covariance matrix is the optimal choice for minimizing such cross-client interference.

### C.1. Proof of Tail-Subspace Decorrelation

**Setup.** Fix a client $k$. Let the single-LoRA residual for client $k$ be

$$R_k = B_k - B_{\mathrm{avg}} \in \mathbb{R}^{d_{\mathrm{out}} \times r},$$

and define the (other-clients) aggregated covariance

$$C_k = \sum_{i \neq k} p_i A_i A_i^\top \quad \text{with} \quad p_i = \frac{N_i}{\sum_{j \neq k} N_j},$$

where $A_i \in \mathbb{R}^{r \times d_{\mathrm{in}}}$ is client $i$'s low-rank down-projection and $N_i$ is its local sample count. Given a rank-$r$ projection $\mathcal{P}_k \in \mathbb{R}^{r \times r}$ with orthonormal columns ($\mathcal{P}_k^\top \mathcal{P}_k = \mathbf{I}_r$), the decorrelated residual is $\widetilde{R}_k = R_k \mathcal{P}_k$.

**Target quantity.** The cross-client interference objective we aim to suppress is

$$
\begin{aligned}
\mathcal{J}_k &\triangleq \sum_{i \neq k} \left\| \widetilde{R}_k A_i \right\|_F^2 \\
&= \sum_{i \neq k} \mathrm{Tr}\left( A_i^\top \widetilde{R}_k^\top \widetilde{R}_k A_i \right) \\
&= \mathrm{Tr}\left( \widetilde{R}_k C_k \widetilde{R}_k^\top \right).
\end{aligned}
$$

Hence, minimizing the target over the choice of $\mathcal{P}_k$ directly corresponds to reducing the representational coupling $\|(B_k - B_{\mathrm{avg}}) A_i\|_F^2$ aggregated over $i \neq k$.

*Proof.* **Goal.** Minimize $\mathcal{J}_k(\mathcal{P}_k) = \sum_{i \neq k} \|\widetilde{R}_k A_i\|_F^2$ over orthonormal $\mathcal{P}_k$.

**Cyclicity.** With $M_k := R_k^\top R_k$,

$$
\begin{aligned}
\mathcal{J}_k(\mathcal{P}_k) &= \mathrm{Tr}\left( R_k \mathcal{P}_k C_k \mathcal{P}_k^\top R_k^\top \right) \\
&= \mathrm{Tr}\left( \mathcal{P}_k^\top M_k \mathcal{P}_k C_k \right).
\end{aligned}
$$

**Isotropy surrogate.** If $M_k \approx \alpha A_k I_r$ with $\alpha A_k > 0$,

$$\mathcal{J}_k(\mathcal{P}_k) \approx \alpha A_k \mathrm{Tr}\left( \mathcal{P}_k^\top C_k \mathcal{P}_k \right), \qquad \text{s.t. } \mathcal{P}_k^\top \mathcal{P}_k = I_r.$$

**Ky Fan / Rayleigh–Ritz.** For symmetric PSD $C_k$,

$$\min_{\mathcal{P}_k^\top \mathcal{P}_k = \mathbf{I}_r} \mathrm{Tr}\left( \mathcal{P}_k^\top C_k \mathcal{P}_k \right) = \sum_{j=1}^r \lambda_{(j)}^\downarrow (C_k),$$

*Table 7*. **Detailed Dataset Description**.

| Dataset | Task | Description | Example |
|---|---|---|---|
| *Textual Scenario* | | | |
| **GLUE** | SST-2 | Sentiment classification of movie reviews (positive/negative). | *The movie was absolutely wonderful! Positive* |
| | QNLI | Question–answer entailment detection. | *Q: What causes rain? A: Water vapor condenses.* |
| | MRPC | Paraphrase identification between two sentences. | *Sentence 1 and Sentence 2 convey the same meaning.* |
| | QQP | Quora question pair classification (duplicate or not). | *Q1: How to lose weight fast? Q2: What are ways to reduce fat quickly?* |
| | RTE | Recognizing textual entailment between premise and hypothesis. | *Premise: Cats are mammals. Hypothesis: Cats give birth to live young.* |
| | MNLI | Multi-genre textual entailment classification. | *A news sentence and a hypothesis for entailment judgment.* |
| *Visual Scenario* | | | |
| **DomainNet** | Quickdraw | Sketch-style drawings of common objects. | |
| | Clipart | Cartoon-like, artistic domain images. | |
| | Painting | Artistic painting samples. | |
| | Sketch | Hand-drawn sketches of objects. | |
| | Real | Real-world photographic images. | |
| | Infograph | Infographic-style visual domain. | |

achieved when $\mathrm{span}(\mathcal{P}_k)$ is the eigenspace of the $r$ smallest eigenvalues of $\mathbf{C}_k$ (tail subspace).

**Conclusion.** Taking $\mathcal{P}_k = \mathrm{TailRank}_r(\mathbf{C}_k)$,

$$\mathcal{J}_k(\mathcal{P}_k) \;\approx\; \alpha \boldsymbol{A_k} \sum_{j=1}^{r} \lambda_{(j)}^{\downarrow}(\mathbf{C}_k),$$

and the positive factor $\alpha \boldsymbol{A_k}$ does not change the minimizer. $\square$

# D. Addtional Experments

## D.1. Datasets Details

We evaluate FedPissa across both textual and visual federated scenarios using two widely adopted benchmarks: GLUE and DomainNet. Table 7 provides a detailed summary of all tasks and example inputs.

**Textual Scenario (GLUE).** The GLUE benchmark consists of six natural language understanding tasks with diverse linguistic properties. SST-2 focuses on binary sentiment classification of movie reviews. QNLI and RTE evaluate sentence–pair entailment, where the model determines whether the hypothesis logically follows the premise. MRPC and QQP address paraphrase identification, requiring recognition of semantic similarity between two sentences or user queries. MNLI is a large-scale multi-genre entailment task involving diverse textual sources. These tasks collectively cover various reasoning and semantic matching abilities crucial for evaluating federated adaptation of language models.

**Visual Scenario (DomainNet).** DomainNet is a large-scale visual benchmark encompassing six heterogeneous visual domains. Quickdraw contains sketch-style drawings with abstract, sparse strokes. Clipart includes cartoon-like, artistic illustrations. Painting consists of artistic paintings with diverse textures and colors. Sketch captures hand-drawn object sketches, while Real contains real-world photographic images. Infograph features infographic-style images with symbolic elements. The large domain shift among these subsets makes DomainNeta challenging testbed for cross-domain personalization in federated vision tasks.

## D.2. Training Process curve

Figure 8 presents the training accuracy curves on the DomainNet benchmark using the ViT-Base backbone. Compared with other pFedLoRA baselines, **FedPissa** exhibits notably faster convergence and achieves a higher final accuracy. This improvement arises from the selective aggregation of adaptive components, which accelerates the global optimization process, and the decorrelated projection mechanism that alleviates cross-domain interference. In contrast, FedLEASE and FedSA-LoRA show slower adaptation due to redundant dual-LoRA structures, while FFA-LoRA underperforms because of its limited learning capacity caused by fixed low-rank mappings. Overall, FedPissa effectively enhances federated fine-tuning efficiency and personalization across heterogeneous visual domains.

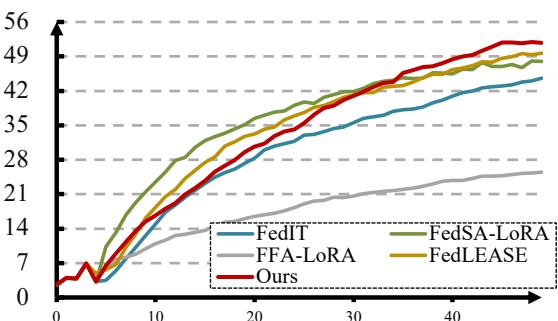

*Figure 8.* Mean accuracy comparison during federated process with counterparts under various heterogeneous degrees. We report the accuracy curves of different methods using the ViT-Base backbone under the feature Non-IID setting. FedPissa demonstrates faster convergence and higher final performance compared to existing pFedLoRA baselines.

## D.3. Lambda $\lambda$ Selection

Figure 9 reports the performance of FedPissa under different decorrelation strengths $\lambda \in \{0.05, 0.1, 0.5, 1.0\}$ on all GLUE and DomainNet tasks. Overall, a moderate decorrelation level yields the most stable gains: on textual tasks, $\lambda$=0.1 consistently matches or surpasses other settings, while overly small or large values occasionally hurt performance on more challenging benchmarks such as RTE and MNLI. For visual domains in DomainNet, a milder setting $\lambda$=0.05 works best, suggesting that strong decorrelation may over-filter domain-specific cues in highly diverse visual distributions. These results indicate that FedPissa is robust to the choice of $\lambda$ in a reasonable range, and that task modality and heterogeneity slightly affect the optimal decorrelation strength.

## D.4. Target Objective

Figure 10 compares three aggregation variants on GLUE: (i) **FedIT**, which performs simple LoRA averaging; (ii) **Agg**

$B$ (FMS), which selectively aggregates the $B$ component; and (iii) **FedPissa**, which further incorporates personalized matrix-level adaptation (PMA). Across all six tasks, Agg-$B$ provides clear improvements over FedIT, confirming that selectively aggregating the adaptive component enhances global optimization and stabilizes convergence. FedPissa consistently achieves the highest accuracy and the fastest convergence, demonstrating the importance of additionally decorrelating personalized updates. The combination of selective matrix aggregation and personalized subspace projection yields the most effective balance between knowledge sharing and client-specific adaptation.

*Table 9.* Effect of client scale under the feature and label non-IID setting with $\beta = 0.5$. We compare the default setting, where each task is distributed to two clients, with a larger-scale setting, where each task is distributed to three clients. The best result under each client-scale setting is highlighted in bold.

| Clients per Task | Methods | QQP | RTE | MNLI | QNLI | Mean |
|---|---|---|---|---|---|---|
| 2 | FedIT | 74.85 | 60.60 | 47.70 | 76.80 | 64.99 |
| | FedLEASE | 76.75 | 55.30 | 57.45 | 82.50 | 68.00 |
| | **FedPissa** | **85.00** | **66.45** | **76.45** | **83.75** | **77.91** |
| 3 | FedIT | 68.90 | 57.68 | 43.80 | 72.43 | 60.70 |
| | FedLEASE | 78.57 | 57.16 | 62.00 | 92.90 | 72.66 |
| | **FedPissa** | **88.00** | **65.42** | **83.33** | **94.70** | **82.86** |

## D.5. Effect of client scale.

We further study the effect of client scale under the feature and label non-IID setting with $\beta = 0.5$. As shown in Table 9, FedPissa achieves the best mean accuracy when each task is distributed to either two or three clients. This indicates that our method remains effective when the number of clients per task increases.

*Table 8.* Performance comparison across six subdomains of the DomainNet dataset. Dual-LoRA methods (e.g., FedDPA, FedLEASE) and Single-LoRA methods (e.g., FedIT, FedSA, FFA-LoRA) are evaluated under the *feature non-IID* scenario. For FedPissa, arrows indicate absolute performance changes relative to FedIT.

| LoRA Type | Methods | Domain Performance on DomainNet ↑ | | | | | | Overall |
|---|---|---|---|---|---|---|---|---|
| | | Quickdraw | Clipart | Painting | Sketch | Real | Infograph | |
| Dual LoRA | FedDPA | 42.80 | 17.80 | 41.50 | 24.50 | 61.50 | 35.80 | 37.32 |
| | FedLEASE | 56.00 | **36.30** | 48.30 | **46.10** | 61.50 | 47.20 | 49.23 |
| Single LoRA | FedIT | 54.20 | 34.70 | 43.80 | 29.20 | 53.60 | 46.90 | 43.73 |
| | FedSA | 57.10 | 24.10 | 46.00 | 42.10 | 69.10 | 46.20 | 47.43 |
| | FFA-LoRA | 30.70 | 7.30 | 29.50 | 13.10 | 43.60 | 26.80 | 25.17 |
| | **FedPissa** | **61.40** ↑7.20 | 36.10 ↑1.40 | **48.50** ↑4.70 | 39.20 ↑10.00 | **71.30** ↑17.70 | **54.30** ↑7.40 | **51.80** ↑8.07 |

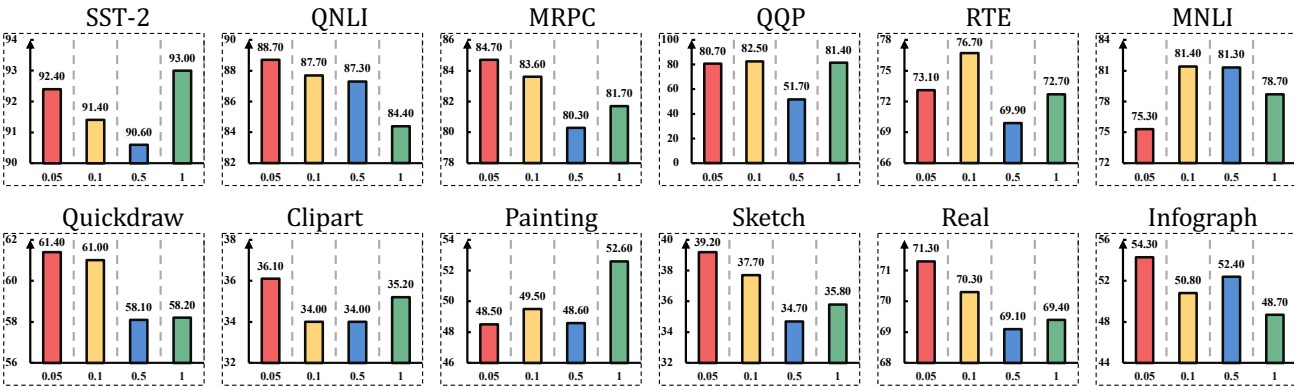

*Figure 9.* Performance of FedPissa across varying $\lambda$ values on GLUE (top) and DomainNet (bottom) benchmarks, illustrating its robustness under different decorrelation strengths.

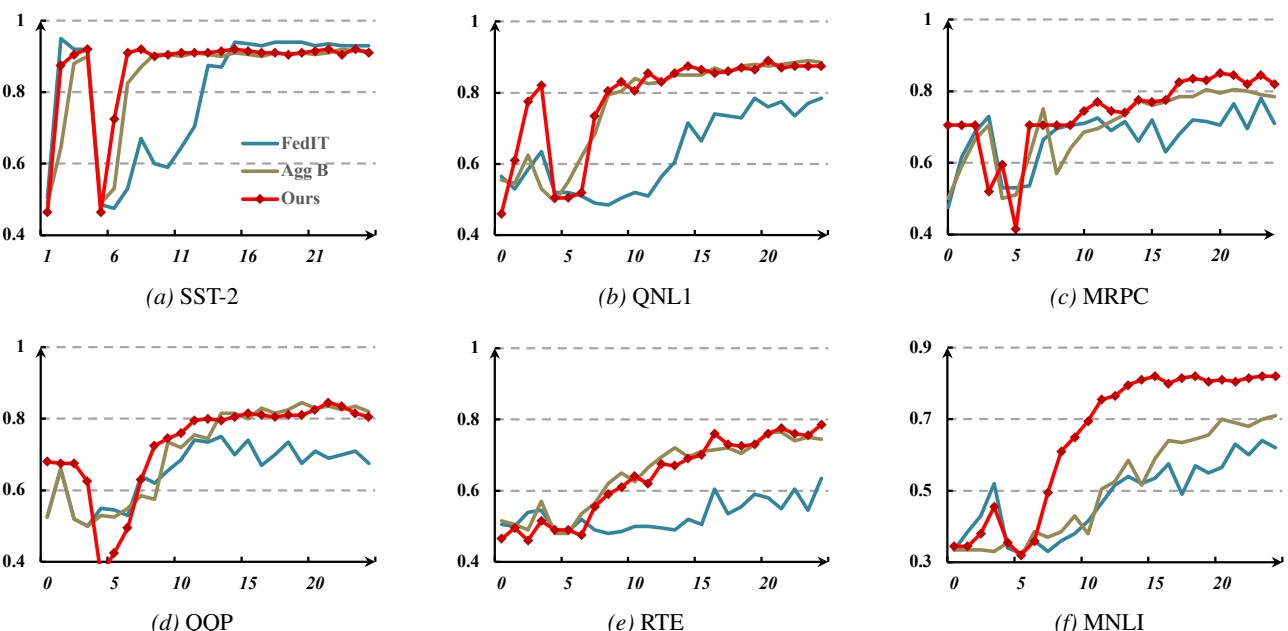

*(a)* SST-2      *(b)* QNL1      *(c)* MRPC

*(d)* QQP      *(e)* RTE      *(f)* MNLI

*Figure 10.* Comparison of different aggregation strategies on GLUE. We compare three variants: (i) FedIT: simple LoRA averaging; (ii) Agg B, *i.e.*, Federated Matrix Selection (FMS); (iii) FedPissa (Federated Matrix Selection(FMS)+Personalized Matrix-level Adaptation(PMA)).

