# OpenReview forum: "FedPissa: Towards Federated Personalized Adaptation of Foundation Models via LoRA Subspace Mapping"
_ICML.cc/2026/Conference — ICML 2026 spotlight_

### Official Review · Reviewer_fgMG · 2026-02-25

**Soundness:** 3
**Presentation:** 4
**Significance:** 3
**Originality:** 4
**Overall Recommendation:** 5
**Confidence:** 5

**Summary:**

FedPissa presents a streamlined approach to personalized federated learning by simplifying the dual-LoRA structure into a single-LoRA design with selective parameter aggregation. The key innovation lies in keeping adapter matrices A local to clients while aggregating only B matrices globally, combined with a subspace decorrelation technique to reduce client interference. While the method shows promising efficiency gains, several implementation details and comparative analyses require further clarification.

**Compliance With Llm Reviewing Policy:**

Affirmed.

**Final Justification:**

My questions and concerns have been sufficiently addressed

**Key Questions For Authors:**

Please see weaknesses above

**Limitations:**

yes

**Strengths And Weaknesses:**

Strengths：
1. The motivation is clear and well-articulated, making the method easy to follow.
2. The proposed method aligns well with the identified motivation, showing a coherent design logic.
3. The experiments are comprehensive and sufficiently support the claimed contributions.

Weaknesses:
1. The paper only evaluates FedPissa with a fixed LoRA rank, but how does the method perform under different rank settings? Would the performance gains hold when using smaller or larger ranks?
2. It would be helpful to include some discussion on security and privacy aspects. For instance, what are the implications when uploading half a LoRA module versus the full module or dual modules? A detailed analysis would strengthen the paper.
3. The experiments cover both NLP and vision tasks, but adding clearer metric descriptions would help readers better understand and interpret the reported results.

---

> ### Author Rebuttal · Authors · 2026-03-29
>
> # W1: Ablation on LoRA Rank.
>
> Thanks for your helpful question. To further examine the effect of LoRA rank, we additionally evaluate FedPissa with a larger rank setting of $r=16$. The results are shown below.
>
> *Table 1: Effect on Larger LoRA Rank ($r=16$).*
>
> |Method|SST-2|QNLI|MRPC|QQP|
> |-|-:|-:|-:|-:|
> |FedAvg|88.90|74.50|75.60|78.10|
> |FedLease|90.00|77.70|76.20|65.00|
> |FedSA|92.70|81.60|82.20|81.80|
> |Ours|**93.50**|**86.30**|**85.60**|**82.30**|
>
> These results show that FedPissa still remains competitive under a larger rank setting. **Although the LoRA rank changes, the subspace mapping remains stable and reduce cross-client interference effectively**. We will add this experiment in the revision.
>
> # W2: Discussion on privacy concern.
>
> Thanks for your valuable suggestion. We would like to clarify that FedPissa focuses on federated personalization. **However, compared with uploading the full LoRA module or dual modules, FedPissa only shares the aggregated $B$ matrix while preserving client-specific information through subspace mapping.** Therefore, it can reduce parameters exposure in some extent while still achieving better performance. We will add this point in our paper.
>
> # W3: Clarification for metric descriptions.
>
> Thanks for your helpful comment. For GLUE, the reported performance is the test accuracy (%) on each task, and the final Mean is the average accuracy across the last 5 rounds and same-task clients. For DomainNet, following common practice in federated domain personalization, we report the classification accuracy (%) on each domain and then average still across the last 5 rounds and same-task clients. We will clarify the evaluation metrics more explicitly in the revised version.

---

> > ### Author Rebuttal · Reviewer_fgMG · 2026-03-31
> >
> > No further questions.

---

> > > ### Author Response · Authors · 2026-04-02
> > >
> > > We thank you for the feedback and for the careful reading. We appreciate the your time and consideration.

---

### Official Review · Reviewer_7rUk · 2026-02-26

**Soundness:** 3
**Presentation:** 3
**Significance:** 3
**Originality:** 3
**Overall Recommendation:** 4
**Confidence:** 4

**Summary:**

This paper proposes FedPissa, a framework for personalized federated learning with LoRA-based fine-tuning of foundation models. The authors identify a key limitation in existing personalized federated LoRA methods: the dual-LoRA architecture used for separating global and local knowledge incurs significant computational overhead and structural redundancy. FedPissa addresses this by introducing a single-LoRA approach with two main components: Federated Matrix Selection, which selectively aggregates only the B matrices across clients while keeping A matrices local, and Personalized Matrix-level Adaptation, which applies decorrelated subspace projection to mitigate cross-client interference.

**Compliance With Llm Reviewing Policy:**

Affirmed.

**Final Justification:**

The authors’ response has fully addressed my concerns, and I will maintain my positive score.

**Key Questions For Authors:**

1.	How sensitive is the method to the number of clients? The experiments use only 6–12 clients, whereas practical federated systems often involve many more.
2.	What happens when you increase the number of local epochs? Does stronger client drift reduce the effectiveness of the decorrelation mechanism?
3.	FedSA argues that A should be globally shared while B remains local. Figure 2 indicates that aggregating B improves convergence, this seems to contradict FedSA's design choice. How do you reconcile this difference in conclusions?

**Limitations:**

yes

**Strengths And Weaknesses:**

Strengths:

1.	The paper is overall well-written and easy to follow. The topic of personalized federated fine-tuning of foundation models is timely.
2.	The empirical observation that LoRA's A and B matrices exhibit different convergence and similarity behaviors is supported by visualization.
3.	The single-LoRA design achieves substantial reductions in computational complexity compared to existing dual LoRA methods while maintaining or improving performance.
4.	The decorrelated subspace projection method is supported by theoretical analysis, showing that tail-eigenspace projection minimizes cross-client interference objective.
5.	Evaluation spans both NLP (GLUE) and vision (DomainNet), which strengthens claims of modality-agnostic applicability.

Weaknesses:

1.	The paper should include a more thorough sensitivity analysis. It is unclear how robust the method is to changes in local epochs or the number of clients.
2.	The label non-IID setting with \beta = 0.1 on GLUE seems quite extreme. For the smaller 2-class/3-class classification datasets in GLUE, this likely leads to highly skewed or near single-label client distributions.
3.	The variance reported in the tables is computed from the last five epochs of a single run. Running multiple experiments with different random seeds would provide a more reliable measure of robustness and statistical significance.
4.	The theoretical argument depends on the assumption M_k \approx \alpha_{A_k} I_r, effectively treating the residual covariance as isotropic.

---

> ### Author Rebuttal · Authors · 2026-03-29
>
> # W1&Q1&Q2: Ablation on local training and clients scale.
> Thanks for your useful suggestion. **To examine the sensitivity of FedPissa to local epochs and client scale**, we add the following experiments.
>
> *Table 1: Effect of local training epochs.*
>
> |Local Epochs|Method  |SST-2|QNLI|MRPC|Mean|
> |-|-|-:|-:|-:|-:|
> |3|FedAvg  |91.00|79.80|70.60|80.47|
> |3|FedLease|91.10|**89.00**|77.50|85.87|
> |3|Ours    |**93.60**|84.40|**82.40**|**86.80**|
> |4|FedAvg  |93.20|83.50|75.70|84.13|
> |4|FedLease|92.60|80.50|79.30|84.13|
> |4|Ours    |**96.50**|**88.50**|**88.00**|**91.00**|
> |5|FedAvg  |92.00|83.20|69.80|81.67|
> |5|FedLease|89.50|75.10|76.70|80.43|
> |5|Ours    |**94.20**|**85.90**|**87.60**|**89.23**|
>
> *Table 2: Effect of client scale.*
> *(each task is distributed to **3 clients**, $\beta=0.5$)*
>
> |Method|QQP|RTE|MNLI|QNLI|Mean|
> |-|-:|-:|-:|-:|-:|
> |FedAvg|66.73|62.32|33.33|71.03|58.35|
> |FedLease|51.83|57.79|**81.43**|50.97|60.51|
> |Ours|**66.87**|**65.01**|52.77|**71.07**|**63.93**|
>
> These results show that FedPissa remains competitive under different local epochs and client scale, suggesting that the proposed LoRA subspace mapping is robust. We will add this point in our paper.
>
> # W2: Discussion about non-IID setting with $\beta=0.1$.
>
> Thank you for this thoughtful comment. We set $\beta=0.1$ as a **challenging heterogeneous setting to evaluate federated personalization under severe label skew.** Under this challenging scenario, FedPissa still remains effective.
>
> Furthermore, we evaluate a milder non-IID setting with $\beta=1.0$. The results are shown below.
>
> *Table 3: Effect of non-IID setting with $\beta=1.0$.*
>
> |Method|SST-2|QNLI|MRPC|QQP|RTE|MNLI|Mean|
> |-|-:|-:|-:|-:|-:|-:|-:|
> |FedAvg|**90.25**|77.60|77.10|78.65|60.43|64.30|74.72|
> |FedLease|68.40|65.65|74.50|78.45|54.86|70.30|68.69|
> |Ours|84.20|**85.00**|**84.10**|**80.00**|**73.33**|**79.50**|**81.02**|
>
> Under a more stable label-skew setting, which is more friendly to small 2/3-class tasks, FedPissa still achieves stable performance. **These label-skew experiments show that aggregating the $B$ matrix with subspace mapping can preserve client-specific personalized knowledge.** We will clarify this point in the revision.
>
> # W3: Clarification for variance estimation.
>
> Thank you for this thoughtful comment. We would like to clarify that the variance reported in our tables is defined differently across settings. **In the feature non-IID setting, the variance is computed from the last five epochs of a single client, mainly reflecting stability. In the feature + label non-IID setting, where the same task is distributed to multiple clients, the variance is computed from the final results of different clients under the same task.**
>
> In the feature + label non-IID setting, heterogeneity is much stronger, and **we care more about whether a method can maintain strong task-level average performance while accommodating different client-specific needs.** Therefore, the variance across clients under the same task is more informative. Under this scenario, FedPissa aggregates $B$ while using subspace mapping to reduce client interference, better preserving client-specific adaptation under the same task.
>
> **To address seed stability, we additionally repeat the experiments with three seeds (1999, 1024, 2026) and report the mean results below.**
>
> *Table 4: Effect on seed stability.*
>
> |Method|MNLI|QNLI|QQP|RTE|Mean|
> |-|-:|-:|-:|-:|-:|
> |FedAvg|47.03|57.36|65.05|54.34|55.95|
> |Ours|**68.14**|**68.32**|**72.19**|**60.20**|**67.21**|
>
> # W4: Clarification for $M_k \approx \alpha_{A_k} I_r$.
>
> Thanks for your thoughtful question. We would like to clarify that
> $$
> M_k = (B_k - B_{\mathrm{avg}})^\top (B_k - B_{\mathrm{avg}})
> $$
> is the second-order structure of the $B$ matrix. **Here, the client-specific residual $B_k - B_{\mathrm{avg}}$ is relative to the average of the other clients.** This is also consistent with $I_r = P_k P_k^\top$, since $P_k$ is computed from
> $$
> C_k = \sum_{i \neq k} p_i A_i A_i^\top,
> $$
> **which also summarizes the shared structure of the other clients.** Therefore, both $M_k$ and $I_r$ are defined with respect to the other-client average, which provides the basis for using $\alpha_{A_k} I_r$ as an isotropic approximation in our analysis. **Furthermore, our assumption is introduced in the same spirit as prior LoRA theory [1, 2], which uses isotropic assumptions to enable more tractable closed-form analysis.**
>
> # Q3: Difference with FedSA.
>
> We would like to clarify that FedSA and FedPissa address personalization under different designs. **FedSA keeps $B$ fully local to preserve personalization, while FedPissa aggregates the shared part of $B$ and restores personalized information through subspace mapping**. We will provide more details in our paper.
>
> [1] One-step full gradient suffices for low-rank fine-tuning, provably and efficiently, ICML'25.
>
> [2] Training-Free Bayesianization for Low-Rank Adapters of Large Language Models, Neurips'25

---

> > ### Author Rebuttal · Reviewer_7rUk · 2026-03-31
> >
> > The authors’ response has fully addressed my concerns, and I will maintain my positive score.

---

> > > ### Author Response · Authors · 2026-04-02
> > >
> > > We sincerely thank you for the positive feedback and for carefully reading our response. We appreciate your recognition that the concerns have been addressed.

---

### Official Review · Reviewer_CtnY · 2026-03-12

**Soundness:** 3
**Presentation:** 3
**Significance:** 3
**Originality:** 3
**Overall Recommendation:** 5
**Confidence:** 5

**Summary:**

This paper rethinks the conventional dual-LoRA architecture and leverages a single-LoRA structure with selective aggregation and subspace decorrelation. By selectively aggregating only the B while keeping A local, and employing a decorrelated projection, FedPissa effectively separates global knowledge from client-specific adaptations. Extensive experiments demonstrate FedPissa achieve better accuracy and lower computational costs compared to other methods.

**Compliance With Llm Reviewing Policy:**

Affirmed.

**Final Justification:**

Please see the Rebuttal Acknowledgement.

**Key Questions For Authors:**

The authors should provide some clarifications to address the concerns raised above.

**Limitations:**

yes

**Strengths And Weaknesses:**

**Strengths：**

(1) The motivation is clear, identifing the computational overhead and cross-client interference issues in dual-LoRA architectures.

(2) The experiments are comprehensive and sufficient, covering multiple nlp and vision benchmarks.

(3) The writing is well-organized with clear technical exposition and effective use of diagrams to illustrate the proposed mechanisms.


**Major Weaknesses:**

(1) I would like to know more about the local training steps and global epoch settings for FedPissa. In my opinion, if the local model is trained with more steps, it may affect the projection matrix P and potentially lead to cross-client interference. It would be better if the authors could provide corresponding ablation studies from this aspect to strengthen the experimental analysis.


(2) I'm hard to understand the communication protocol because I don't know what exactly each client uploads and downloads during the federated learning process.


(3) As I know, FFA-LoRA is also a matrix $B$ aggregation, similar with this paper. What's the difference between FedPissa and FFA-LoRA. And I want to know the key that results the performance difference. However, the author seems to lack the enough details about this.

(4) Figure 2 shows some quit different patterns where Agg A and Agg B appear to have opposite trends. The similarity matrices for Agg B show reversal patterns but author do not explain.

**Minor Weaknesses：**

(1) “Fig.” and “Figure” has some errors. The paper use “Fig. 4” in P7-377, while using “Figure 1” in P2-75.

(2) The model name is inconsistently capitalized. “Roberta” in P2-81 is wrong, which can confuse readers.

---

> ### Author Rebuttal · Authors · 2026-03-29
>
> # W1: Ablation on local training and global communication settings.
> Thanks for your insightful comment. Our default setting is 2 local iterations per communication round and 25 global rounds.
> We would like to clarify that the computation of **$P_k$ in FedPissa is mainly motivated by the cross-client heterogeneity knowledge, which commonly exists across global iterations in pFedLoRA.**
> Specifically, $P_k$ is computed as $P_k = \mathrm{TailRank}_r(C_k)$, where $C_k$ is constructed from other heterogeneous clients. **This design allows $P_k$ to capture the isolated subspace across other clients and project the personalized knowledge into this subspace, reducing cross-client interference.** We additionally provide ablations on more local epochs and global rounds:
>
> *Table 1: Effect of local training epochs.*
>
> | Local Epochs | Method   | SST-2 | QNLI | MRPC | Mean |
> |---|---|---:|---:|---:|---:|
> | 3 | FedAvg   | 91.00 | 79.80 | 70.60 | 80.47 |
> | 3 | FedLease | 91.10 | **89.00** | 77.50 | 85.87 |
> | 3 | Ours     | **93.60** | 84.40 | **82.40** | **86.80** |
> | 4 | FedAvg   | 93.20 | 83.50 | 75.70 | 84.13 |
> | 4 | FedLease | 92.60 | 80.50 | 79.30 | 84.13 |
> | 4 | Ours     | **96.50** | **88.50** | **88.00** | **91.00** |
> | 5 | FedAvg   | 92.00 | 83.20 | 69.80 | 81.67 |
> | 5 | FedLease | 89.50 | 75.10 | 76.70 | 80.43 |
> | 5 | Ours     | **94.20** | **85.90** | **87.60** | **89.23** |
>
> *Table 2: Effect of global communication rounds.*
>
> | Global Rounds | Method   | SST-2 | QNLI | MRPC | Mean |
> |---|---|---:|---:|---:|---:|
> | 30 | FedAvg   | 90.40 | 75.20 | 72.80 | 79.47 |
> | 30 | FedLease | 91.80 | 80.60 | 77.10 | 83.17 |
> | 30 | Ours     | **94.80** | **85.30** | **85.40** | **88.50** |
> | 35 | FedAvg   | **94.90** | 78.50 | 72.10 | 81.83 |
> | 35 | FedLease | 92.00 | **88.10** | 83.10 | 87.73 |
> | 35 | Ours     | 94.10 | 87.50 | **86.00** | **89.20** |
>
> FedPissa consistently remains competitive under different settings, which suggests that the proposed subspace mapping  mechanism is stable. We will add this clarification in the revision.
>
> # W2: Clarification about communication protocol.
> We would like to clarify the detail communication protocol. In each communication round, **each client downloads only $B_{\text{agg}}^t$ from the server, while keeping its own $A_k$ local for personalized adaptation**. After local training, each client **uploads its updated $A_k^{t+1}$ and $B_k^{t+1}$ to the server.** The server then uses all uploaded parameters to compute the next aggregated $B_{\text{agg}}^{t+1}$ and the personalized projection matrices $P_k$, and only $B_{\text{agg}}^{t+1}$ is broadcast back to clients. We will add more details into Algorithm 1 in paper.
>
> # W3: Further discussion on the difference between FedPissa and FFA-LoRA
> Thanks for your important question. We have briefly introduced FFA-LoRA in Line 321 in paper, and we further clarify the difference here.
>
> **For FFA-LoRA, the global update follows a simple aggregation scheme on the trainable LoRA factor while freezing the other:**
> $$
> \text{Freezing } A, \qquad
> B_{\mathrm{g}} = \sum_{k=1}^{K} p_k B_k.
> $$
>
> **In contrast, FedPissa keeps $A$ trainable and performs aggregation with personalized subspace correction:**
> $$
> \text{Training } A, \qquad
> B_{\mathrm{agg}} = B_{\mathrm{avg}} + \lambda \sum_{k=1}^{K} (B_k - B_{\mathrm{avg}}) P_k.
> $$
>
> Therefore, the key difference is that **FFA-LoRA mainly relies on direct parameter aggregation**, while **FedPissa further models cross-client heterogeneity through the client-specific projection matrix $P_k$**. This personalized subspace mapping helps preserve shared knowledge while reducing interference from heterogeneous client updates, resulting in FedPissa better performance. We will clarify this distinction more explicitly in the revision.
>
> # W4: Further discussion on Figure 2.
>
> Thanks for your careful observation. We briefly discuss Figure 2 in Line 179 of the paper and further clarify it here. The opposite trends of ***Agg A and Agg B*** come from their different roles in LoRA: **$A$ mainly captures shared knowledge across clients, while $B$ mainly reflects task-specific adaptation**.
> As training proceeds, directly aggregating $B$ gradually makes the personalized information increasingly pulled toward shared directions, causing the reversal pattern. **However, FedPissa performs personalized subspace mapping on $B$ to reduce interference from clients updates, making better performance across different tasks.** We will add further explanation in the revision.
>
> # W5&W6: Writing Mistakes
> We sincerely appreciate your careful reading, and we will carefully check the paper and revise writing mistakes.

---

> > ### Author Rebuttal · Reviewer_CtnY · 2026-04-02
> >
> > I appreciate the authors’ effort in the rebuttal. Their responses have addressed my concerns, and I retain my initial positive rating.

---

> > > ### Author Response · Authors · 2026-04-07
> > >
> > > We thank you for the positive feedback and for the careful reading. We are glad that our responses have addressed the concerns, and we sincerely appreciate your support.

---

### Official Review · Reviewer_qciS · 2026-03-13

**Soundness:** 3
**Presentation:** 4
**Significance:** 3
**Originality:** 4
**Overall Recommendation:** 4
**Confidence:** 4

**Summary:**

This paper proposes FedPissa, a single-LoRA personalized federated fine-tuning framework that replaces dual-LoRA designs with selective aggregation and subspace decorrelation to reduce redundancy and cross-client interference. Specifically, FedPissa aggregates the LoRA $B$ matrices while keeping $A$ local, and further refines the aggregated update via a decorrelated tail-subspace projection to preserve client-specific residuals.

**Compliance With Llm Reviewing Policy:**

Affirmed.

**Final Justification:**

Following the authors' response, I am more convinced of the value and correctness of this work. The rebuttal addresses my main concerns and provides enough clarification for me to retain a positive overall evaluation.

**Key Questions For Authors:**

See the weaknesses. I hope the author can carefully check the paper grammer.

**Limitations:**

Yes.

**Strengths And Weaknesses:**

Strengths：

First, this method is intuitive, easy to implement, and directly targets. cross-client interference.

Second, this algorithm is novelty and effectively solve its motivation based on dual-lora architecture.

Third, the experiments span both NLP and Vision tasks, showing consistent. personalization gains.


Weaknesses:

First, all Tables results are shown as averages. However, in some scenarios (label non-iid), does all clients contains different tasks? If do, how does the author test the label-level heterogeneous performance? If not, I think it neccessary to give a detailed discription of each clients performance.

Second, some settings are under-explored, such as local training and global communication setting. It's unclear whether the method remains fair under different training and communication setup.

Third, Fig. 2 caption is grammatically awkward and unclear (e.g., “brings slowly personalization”, “faster converge”), which hurts readability. it should be revised to standard phrasing like “leads to slower personalization” and “faster convergence.”

---

> ### Author Rebuttal · Authors · 2026-03-29
>
> # W1: Clarification for label non-iid setting.
> Thank your for raising this point. **We need to clarify that under label non-iid setting, task heterogeneity and label heterogeneity coexist (line 342 in paper).**
>
> **Specifically, we use 12 clients, with every two clients assigned to one task, giving 6 tasks in total. Within each task, the two clients are split by a Dirichlet distribution.** The final result is the average of the two clients, further averaged over the last 5 rounds. Thanks for your detailed reading and we will also add more detailed discription in the revision.
>
> # W2: Ablation on local training and global communication settings.
>
> Thank your for this helpful suggestion. **To further examine the effect of local training and global communication settings, we additionally evaluate our method under different numbers of local epochs and global rounds.** The results are shown below.
>
> *Table 1: Effect of local training epochs.*
>
> | Local Epochs | Method   | SST-2 | QNLI | MRPC | Mean |
> |---|---|---:|---:|---:|---:|
> | 3 | FedAvg   | 91.00 | 79.80 | 70.60 | 80.47 |
> | 3 | FedLease | 91.10 | **89.00** | 77.50 | 85.87 |
> | 3 | Ours     | **93.60** | 84.40 | **82.40** | **86.80** |
> | 4 | FedAvg   | 93.20 | 83.50 | 75.70 | 84.13 |
> | 4 | FedLease | 92.60 | 80.50 | 79.30 | 84.13 |
> | 4 | Ours     | **96.50** | **88.50** | **88.00** | **91.00** |
> | 5 | FedAvg   | 92.00 | 83.20 | 69.80 | 81.67 |
> | 5 | FedLease | 89.50 | 75.10 | 76.70 | 80.43 |
> | 5 | Ours     | **94.20** | **85.90** | **87.60** | **89.23** |
>
> *Table 2: Effect of global communication rounds.*
>
> | Global Rounds | Method   | SST-2 | QNLI | MRPC | Mean |
> |---|---|---:|---:|---:|---:|
> | 30 | FedAvg   | 90.40 | 75.20 | 72.80 | 79.47 |
> | 30 | FedLease | 91.80 | 80.60 | 77.10 | 83.17 |
> | 30 | Ours     | **94.80** | **85.30** | **85.40** | **88.50** |
> | 35 | FedAvg   | 94.90 | 78.50 | 72.10 | 81.83 |
> | 35 | FedLease | 92.00 | **88.10** | 83.10 | 87.73 |
> | 35 | Ours     | 94.10 | 87.50 | **86.00** | **89.20** |
>
> We can see that FedPissa consistently remains competitive under different settings, which suggests that the proposed LoRA subspace mapping is stable and that $P_k$ does not rely on a specific training budget.
>
> # W3: Concerns about the wording of the Fig. 2 caption
>
> Thank your for pointing this out. We will revise it in our paper.

---

> > ### Author Rebuttal · Reviewer_qciS · 2026-04-04
> >
> > Appreciate the reviewer’s detailed explanation, which resolves my concerns.

---

> > > ### Author Response · Authors · 2026-04-07
> > >
> > > We appreciate your careful reading and are glad that our clarification resolves the concerns.

---

### Decision · Program_Chairs · 2026-04-30

**Decision:**

Accept (spotlight)

**Comment:**

This paper proposes a single-LoRA personalized federated fine-tuning framework that replaces dual-LoRA designs with selective aggregation and subspace decorrelation to reduce redundancy and cross-client interference. This method is intuitive, easy to implement, and directly targets. cross-client interference.